# Differentially Private $k$-Means with Constant Multiplicative Error

**Haim Kaplan**
Tel Aviv University and Google
haimk@post.tau.ac.il

**Uri Stemmer**[*]
Ben-Gurion University
u@uri.co.il

## Abstract

We design new differentially private algorithms for the Euclidean $k$-means problem, both in the *centralized model* and in the *local model* of differential privacy. In both models, our algorithms achieve significantly improved error guarantees than the previous state-of-the-art. In addition, in the local model, our algorithm significantly reduces the number of interaction rounds.

Although the problem has been widely studied in the context of differential privacy, all of the existing constructions achieve only super constant approximation factors. We present—for the first time—efficient private algorithms for the problem with *constant* multiplicative error. Furthermore, we show how to modify our algorithms so they compute *private coresets* for $k$-means clustering in both models.

## 1 Introduction

Clustering, and in particular center based clustering, are central problems in unsupervised learning. Several cost objectives have been intensively studied for center based clustering, such as minimizing the sum or the maximum of the distances of the input points to the centers. Most often the data is embedded in Euclidean space and the distances that we work with are Euclidean distances. In particular, probably the most studied center based clustering problem is the *Euclidean $k$-means problem*. In this problem we are given a set of $n$ input points in $\mathbb{R}^d$ and our goal is to find $k$ centers that minimize the sum of squared distances between each input point to its nearest center.[2] When privacy is not a concern one usually solves this problem by running Lloyd's algorithm [25] initialized by $k$-means++ [4]. This produces $k$-centers of cost that is no worse than $O(\log k)$ times the cost of the optimal centers and typically much lower in practice.

The huge applicability of $k$-means clustering, together with the increasing awareness and demand for user privacy, motivated the study of *privacy preserving $k$-means algorithms*. It is especially desirable to achieve *differential privacy* [12], a privacy notion which has been widely adopted by the academic community as well as big corporations like Google, Apple, and Microsoft. Indeed, constructions of differentially private $k$-means algorithms have received a lot of attention over the last 14 years [8, 28, 14, 18, 27, 33, 31, 32, 17, 6, 29, 21]. In this work we design new differentially private $k$-means algorithms, both for the centralized model (where a trusted curator collects the sensitive information and analyzes it with differential privacy) and for the local model (where each respondent randomizes her answers to the data curator to protect her privacy). In both models, our algorithms offer significant improvements over the previous state-of-the-art.

---

[*]Work done while the second author was a postdoctoral researcher at the Weizmann Institute of Science, supported by a Koshland fellowship, and by the Israel Science Foundation (grants 950/16 and 5219/17).

[2]The sum of squares is nice to work with since we do not have to compute square roots. Furthermore, for a given cluster its center of mass is the minimizer of the sum of the squared distances. These properties make $k$-means to be the favorite cost objective for center based clustering.

| Reference | Multiplicative Error | Additive Error |
|:---:|:---:|:---:|
| Feldman et al. (2009) [14] | $O(\sqrt{d})$ | $\tilde{O}\left((kd)^{2d}\right)$ |
| Nock et al. (2016) [31] | $O(\log k)$ | $O\left(n/\log^2 n\right)$ |
| Feldman et al. (2017) [17] | $O(k \log n)$ | $\tilde{O}\left(\sqrt{d} \cdot k^{1.5}\right)$ |
| Balcan et al. (2017) [6] | $O(\log^3 n)$ | $\tilde{O}\left(d + k^2\right)$ |
| Nissim and Stemmer (2018) [29] | $O(k)$ | $\tilde{O}\left(d^{0.51} \cdot k^{1.51}\right)$ |
| **This work** | $\boldsymbol{O(1)}$ | $\boldsymbol{\tilde{O}\left(k^{1.01} \cdot d^{0.51} + k^{1.5}\right)}$ |

**Table 1**: Private algorithms for $k$-means. Here $n$ is the number of input points, $k$ is the number of centers, and $d$ is the dimension. For simplicity, we assume that input points come from the unit ball, and omit the dependency in $\varepsilon$, as well as logarithmic factors in $k, n, d, \beta, \delta$, from the additive error.

Before describing our new results, we define our setting more precisely. Consider an input database $S = (x_1, \ldots, x_n) \in (\mathbb{R}^d)^n$ containing $n$ points in $\mathbb{R}^d$, where every point $x_i \in S$ is the (sensitive) information of one individual. The goal is to identify a set of $k$ *centers* $C = \{c_1, \ldots, c_k\}$ in $\mathbb{R}^d$ approximately minimizing the following quantity, referred to as the *cost* of the centers

$$\text{cost}_S(C) = \sum_{i=1}^{n} \min_{j \in [k]} \|x_i - c_j\|_2^2.$$

The privacy requirement is that the output of our algorithm (the set of centers) does not reveal information that is specific to any single individual. Formally,

**Definition 1.1** ([12]). *A randomized algorithm $\mathcal{A} : X^n \to Y$ is $(\varepsilon, \delta)$ differentially private if for every two databases $S, S' \in X^n$ that differ in one point, and every set $T \subseteq Y$, we have $\Pr[\mathcal{A}(S) \in T] \leq e^{\varepsilon} \cdot \Pr[\mathcal{A}(S') \in T] + \delta$.*

Combining the utility and privacy requirements, we are seeking for a computationally efficient differentially private algorithm that identifies a set of $k$ centers $C$ such that w.h.p. $\text{cost}_S(C) \leq \gamma \cdot \text{OPT}_S + \eta$, where $\text{OPT}_S$ is the optimal cost. We want $\gamma$ and $\eta$ to be as small as possible, as a function of the number of input points $n$, the dimension $d$, the number of centers $k$, the failure probability $\beta$, and the privacy parameters $\varepsilon, \delta$.

We remark that a direct consequence of the definition of differential privacy is that, unlike in the non-private literature, every private algorithm for this problem must have additive error $\eta > 0$. In fact, if all points reside with the $d$-dimensional ball, $\mathcal{B}(0, \Lambda)$, of radius $\Lambda$ around the origin (as we assume in this paper) then $\eta$ must be at least $\Lambda^2$. To see this, consider $k + 1$ locations $p_1, \ldots, p_{k+1}$ at pairwise distances $\Lambda$, and consider the following two neighboring datasets. The first dataset $S_1$ contains $n - k + 1$ copies of $p_1$, and (one copy of) $p_2, \ldots, p_k$. The second dataset $S_2$ is obtained from $S_1$ by replacing $p_k$ with $p_{k+1}$. Since in both cases there are only $k$ distinct input points, the optimal cost for each of these datasets is zero. On the other hand, by the constraint of differential privacy, the set of centers we compute essentially cannot be affected by this change. Therefore we must have expected error of $\Omega(\Lambda^2)$ on at least one of these inputs. To simplify the presentation we assume that $\Lambda = 1$ in rest of the introduction.

Traditionally, in the non-private literature, the goal is to minimize the multiplicative error $\gamma$, with the current state-of-the-art (non-private) algorithm achieving multiplicative error of $\gamma = 6.357$ (with no additive error) [2]. In contrast, in spite of the long line of works on private $k$-means [8, 28, 14, 18, 27, 33, 31, 32, 17, 6, 29, 21], all of the existing polynomial time private algorithms for the problem obtained only a super constant multiplicative error. We present the first polynomial time differentially private algorithm for the Euclidean $k$-means problem with constant multiplicative error, while essentially keeping the additive error the same as in previous state-of-the-art results. See Table 1 for a comparison.

## 1.1 Locally private $k$-means

In the local model of differential privacy (LDP), there are $n$ users and an untrusted server. Each user $i$ is holding a private input item $x_i$ (a point in $\mathbb{R}^d$ in our case), and the server's goal is to compute

some function of the inputs (approximate the $k$-means in our case). However, in this model, the users do not send their data as is to the server. Instead, every user randomizes her data locally, and sends a differentially private report to the server, who aggregates all the reports. Informally, the privacy requirement is that the input of user $i$ has almost no effect on the distribution of the messages that user $i$ sends to the server. This is the model used by Apple, Google, and Microsoft in practice to ensure that private data never reaches their servers in the clear.

With increasing demand from the industry, the local model of differential privacy is now becoming more and more popular. Nevertheless, the only currently available $k$-means algorithm in this model (with provable utility guarantees) is that of Nissim and Stemmer [29], with $O(k)$ multiplicative error. We present a new LDP algorithm for the $k$-means achieving *constant* multiplicative error. In addition, the protocol of [29] requires $O(k \log n)$ rounds of interactions between the server and the users, whereas our protocol uses only $O(1)$ such rounds.

## 1.2 Classical algorithms are far from being private

We highlight some of the challenges that arise when trying to construct private variants for existing (non-private) algorithms. Recall for example the classical (non-private) Lloyd's algorithm, where in every iteration the input points are grouped by their proximity to the current centers, and the points in every group are averaged to obtain the centers for the next round. One barrier for constructing a private analogue of this algorithm is that, with differential privacy, the privacy parameters deteriorate with number of (private) computations that we apply to the dataset. So, even if we were able to construct a private analogue for every single iteration, our approximation guarantees would not necessarily improve with every iteration. In more details, composition theorems for differential privacy [13] allow for applying $O(n^2)$ private computations before exhausting the privacy budget completely. Lloyd's algorithm, however, might perform a much larger number of iterations (exponential in $n$ in worst case). Even the bounds on its smoothed complexity are much larger than $n^2$ (currently $\approx n^{32}$ is known [3]). In addition, classical techniques for reducing the number of iterations often involve computations which are highly sensitive to a change of a small number of input points. For example, recall that in $k$-means++ [4] the initial $k$ centers (with which Lloyd's algorithm is typically initiated) are chosen *from the data points themselves*, an operation which cannot be applied as is when the data points are private.

These challenges are reflected in the recent work of Nock et al. [31], who constructed a private variant for the $k$-means++ algorithm. While their private algorithm achieves a relatively low multiplicative error of $O(\log k)$, their additive error is $\tilde{O}(n)$. In this work we are aiming for additive error at most polylogarithmic in $n$. Note that having additive error of $n$ is meaningless, since if points come from the unit ball then *every* choice of $k$ centers have error at most $O(n)$.

## 1.3 On the evolution of private $k$-means algorithms

The starting point of our work is the observation that by combining ideas from three previous works [18, 6, 29] we can obtain a differentially private $k$-means algorithm (in the centralized model) with constant multiplicative error, but with a relatively large additive error which is polynomial in $n$ (as we will see in Section 1.4). Most of our technical efforts (in the centralized model) are devoted to reducing the additive error while keeping the multiplicative error constant. We now describe the results of [18, 6, 29].

Gupta et al. [18] constructed a private variant for the classical local search heuristic [5, 24] for $k$-medians and $k$-means. In this local search heuristic, we start with an arbitrary choice of $k$ centers, and then proceed in iterations, where in every iteration we replace one of our current centers with a new one, so as to reduce the $k$-means cost. Gupta et al. [18] constructed a private variant of the local search heuristic by using the (generally inefficient) exponential mechanism of McSherry and Talwar [26] in order to privately choose a replacement center in every step. While the algorithm of Gupta et al. [18] obtains superb approximation guarantees[3], its runtime is exponential in the representation length of domain elements. Specifically, it is designed for a *discrete* version of the problem, in which centers come from a *finite* set $Y$, and the runtime of their algorithm is at least linear in $|Y|$. In particular, when applying their algorithm to the Euclidean space, one must first *discretize* the space of possible centers, and account for the error introduced by this discretization. For example,

Gupta et al. mentions that one can take $Y$ to be a discretization net of the unit $d$-dimensional ball. However, to ensure small discretization error, such a net would need to be of size $|Y| \approx n^d$, and hence, would result in an inefficient algorithm (since the runtime is linear in $|Y|$).

Balcan et al. [6] suggested the following strategy in order to adopt the techniques of Gupta et al. [18] to the Euclidean space while maintaining efficiency. Instead of having a fixed (data independent) discretization of the unit ball, Balcan et al. suggested to first identify (in a differentially private manner) a small set $Y \subseteq \mathbb{R}^d$ of *candidate centers* such that $Y$ contains a subset of $k$ candidate centers with low $k$-means cost. Then, apply the techniques of Gupta et al. in order to choose $k$ centers from $Y$. If $|Y| = \text{poly}(n)$, then the resulting algorithm would be efficient. As the algorithm of Gupta et al. has very good approximation guarantees, the bottleneck for the approximation error in the algorithm of Balcan et al. is in the construction of $Y$. Namely, the overall error is dominated by the error of the best choice of $k$ centers out of $Y$ (compared to the cost of the best choice of $k$ centers from $\mathbb{R}^d$). At first glance, this might seem easy to achieve, since for non-private $k$-means, one can simply take the input points themselves as the set of candidate centers (this is of size $n$ and has an error of at most 2 compared to centers from $\mathbb{R}^d$). However, for private $k$-means clustering, this is not possible – the centers cannot be a subset of the input points, because otherwise, removing a point may significantly change the computed centers.

Balcan et al. then constructed a differentially private algorithm for identifying a set of candidate centers $Y$ based on the Johnson–Lindenstrauss transform [23]. However, their construction gives a set of candidate centers such that the best choice of $k$ centers from these candidates is only guaranteed to have a multiplicative error of $O(\log^3 n)$, leading to a private $k$-means algorithm with $O(\log^3 n)$ multiplicative error.

A different approach to obtain a good $k$-means clustering privately is via algorithms for the *1-cluster problem*, where given a set of $n$ input points in $\mathbb{R}^d$ and a parameter $t \leq n$, the goal is to identify a ball of the smallest radius that encloses at least $t$ of the input points. It was shown by Feldman et al. [17] that the Euclidean $k$-means problem can be reduced to the 1-cluster problem, by iterating the 1-cluster algorithm multiple times to find several balls that cover most of the data points. Feldman et al. then applied their reduction to the private 1-cluster algorithm of [30], and obtained a private $k$-means algorithm with multiplicative error $O(k \log n)$. Following that work, Nissim and Stemmer [29] presented an improved algorithm for the 1-cluster problem which, when combined with the reduction of Feldman et al., gives a private $k$-means algorithm with multiplicative error $O(k)$.

## 1.4 Our techniques

Let $S \in (\mathbb{R}^d)^n$ be an input database and let $u_1^*, \ldots, u_k^* \in \mathbb{R}^d$ denote an optimal set of centers for $S$. We use $S_j^* \subseteq S$ to denote the cluster induced by $u_j^*$, i.e., $S_j^* = \{x \in S : j = \text{argmin}_\ell \|x - u_\ell^*\|\}$.

We observe that the techniques that Nissim and Stemmer [29] applied to the 1-cluster problem can be extended to privately identify a set of candidate centers $Y$ that "captures" every "big enough" cluster $j$. Informally, let $j$ be such that $|S_j^*| \geq n^a$ (for some constant $a > 0$). We will construct a set of candidate centers $Y$ such that there is a candidate center $y_j \in Y$ that is "close enough" to the optimal center $u_j^*$, in the sense that the cost of $y_j$ w.r.t. $S_j^*$ is at most a constant times bigger than the cost of $u_j^*$. That is, $\text{cost}_{S_j^*}(\{y_j\}) = O\left(\text{cost}_{S_j^*}(\{u_j^*\})\right)$. By simply ignoring clusters of smaller sizes, this means that $Y$ contains a subset $D$ of $k$ candidate centers such that $\text{cost}_S(D) \leq O(1) \cdot \text{OPT}_S + k \cdot n^a$.

There are *two* reasons for the $\text{poly}(n)$ additive error incurred here. First, this technique effectively ignores every cluster of size less than $n^a$, and we pay $n^a$ additive error for every such cluster. Second, this technique only succeeds with polynomially small probability, and boosting the confidence using repetitions causes the privacy parameters to degrade.

We show that it is possible to boost the success probability of the above strategy without degrading the privacy parameters. To that end, we apply the repetitions to disjoint samples of the input points, and show that the sampling process will not incur a $\text{poly}(n)$ error. In order to "capture" smaller clusters, we apply the above strategy repeatedly, where in every iteration we exclude from the computation the closest input points to the set of centers that we have already identified. We show that this technique allows to "capture" much smaller clusters. By combining this with the techniques of Balcan et al. and Gupta et al. for privately choosing $k$ centers out of $Y$, we get our new construction for $k$-means in the centralized model of differential privacy (see Table 1).

**A construction for the local model.** Recall that the algorithm of Gupta et al. (the private variant of the local search) applies the exponential mechanism of McSherry and Talwar [26] in order to privately choose a replacement center in every step. This use of the exponential mechanism is tailored to the centralized model, and it is not clear if the algorithm of Gupta et al. can be implemented in the local model. In addition, since the local search algorithm is iterative with a relatively large number of iterations (roughly $k \log n$ iterations), a local implementation of it, if exists, may have a large number of rounds of interactions between the users and the untrusted server.

To overcome these challenges, in our locally private algorithm for $k$-means we first identify a set of candidate centers $Y$ (in a similar way to the centralized construction). Afterwards, we estimate the *weight* of every candidate center, where the *weight* of a candidate center $y$ is the number of input points $x \in S$ s.t. $y$ is the nearest candidate center to $x$. We show that the weighted set of candidate centers can be post-processed to obtain an approximation to the $k$-means of the input points. In order to estimate the weights we define a natural extension of the well-studied heavy-hitters problem under LDP, which reduces our incurred error.

**Private coresets.** A *coreset* [1] of set of input points $S$ is a small (weighted) set of points $P$ that captures some geometric properties of $S$. Coresets can be used to speed up computations, since if the coreset $P$ is much smaller than $S$, then optimization problems can be solved much faster by running algorithms on $P$ instead of $S$. In the context of $k$-means, the geometric property that we want $P$ to preserve is the $k$-means cost of *every* possible set of centers. That is, for every set of $k$ centers $D \subseteq \mathbb{R}^d$ we want that $\text{cost}_P(D) \approx \text{cost}_S(D)$ (where in $\text{cost}_P(D)$ we multiply each distance by the weight of the corresponding point). Coresets for $k$-means and $k$-medians have been the subject of many recent papers, such as [10, 16, 19, 20, 7, 11, 15]. *Private* coresets for $k$-means and $k$-medians have been considered in [14] and in [17]. We show that our techniques result in new constructions for private coresets for $k$-means and $k$-medians, both for the centralized and for the local model of differential privacy. In the local model, this results in the *first* private coreset scheme with provable utility guarantees. In the centralized model, our new construction achieves significantly improved error rates over the previous state-of-the-art. We omit our results for private coresets due to space restrictions. See the full version of this work for more details.

## 2 Preliminaries from [18, 6]

As we described in the introduction, we use a private variant of the local search algorithm by Gupta et al. and Balcan et al. We now state its guarantees. Let $Y \subseteq \mathbb{R}^d$ be our precomputed set of *candidate centers*. Given a set of points $S \in (\mathbb{R}^d)^n$ consider the task of identifying a subset $C \subseteq Y$ of size $k$ with the lowest possible cost. That is, instead of searching for $k$ centers in $\mathbb{R}^d$, we are searching for $k$ centers in $Y$, and our runtime is allowed to depend polynomially on $|Y|$. We write $\text{OPT}_S(Y)$ to denote the lowest possible cost of $k$ centers from $Y$. That is, $\text{OPT}_S(Y) = \min_{C \subseteq Y, |C|=k} \{\text{cost}_S(C)\}$. Recall that we denote the lowest cost of $k$ centers out of $\mathbb{R}^d$ as $\text{OPT}_S$, i.e., $\text{OPT}_S = \text{OPT}_S(\mathbb{R}^d)$.

**Theorem 2.1** ([18, 6]). *Let $\beta, \varepsilon, \delta > 0$ and $k \in \mathbb{N}$, and let $Y \subseteq \mathbb{R}^d$ be a finite set of centers. There exists an $(\varepsilon, \delta)$-differentially private algorithm that takes a database $S$ containing $n$ points from the $d$-dimensional ball $\mathcal{B}(0, \Lambda)$, and outputs a subset $D \subseteq Y$ of size $|D| = k$ s.t. with probability at least $(1 - \beta)$ we have that*

$$\text{cost}_S(D) \leq O(1) \cdot \text{OPT}_S(Y) + O\left(\frac{k^{1.5}\Lambda^2}{\varepsilon} \log\left(\frac{n|Y|}{\beta}\right) \sqrt{\log(n) \cdot \log\left(\frac{1}{\delta}\right)}\right).$$

In light of Theorem 2.1, in order to privately identify an approximation to the $k$-means of the input set $S$, it suffices to privately identify a set of candidate centers $Y \subseteq \mathbb{R}^d$ such that $|Y| = \text{poly}(n)$, and in addition, $Y$ contains a subset with low $k$-means cost (that is $\text{OPT}_S(Y)$ is comparable to $\text{OPT}_S$).

We remark that $Y$ must be computed using a differentially private algorithm, and that in particular, taking $Y = S$ will *not* lead to a differentially private algorithm (even though $Y = S$ is an excellent set of candidate centers in terms of utility). To see this, let us denote the algorithm from Theorem 2.1 as $\mathcal{A}$. Its inputs are the database $S$ and the set of candidate centers $Y$, and the differential privacy guarantee is only with respect to the database $S$. In other words, for every fixed set $Y$, the algorithm

$\mathcal{A}_Y(S) = \mathcal{A}(S, Y)$ is differentially private as a function of $S$. Known composition theorems for differential privacy [13] show that for every differentially private algorithm $\mathcal{B}$ that takes a database $S$ and outputs a set of centers $Y$, we have that the composition $\mathcal{A}(S, \mathcal{B}(S))$ satisfies differential privacy. On the other hand, there is no guarantee that $\mathcal{A}(S, S)$ is differentially private, and in general it is not.

## 3 Private $k$-means – the centralized setting

In this section we present some of the components of our algorithm for approximating the $k$-means in the centralized model of differential privacy. All of the missing details appear in the full version of this work, as well as our algorithm for the local model, and our construction of a private coreset.

Consider an input database $S$, and let $u_1^*, \ldots, u_k^* \in \mathbb{R}^d$ denote an optimal set of $k$ centers for $S$. Our starting point is the observation that, extending the techniques of Nissim and Stemmer [29], we can identify a set of candidate centers that contains a "close enough" candidate center to every optimal center $u_j^*$, provided that the optimal cluster induced by $u_j^*$ is "big enough". We call this algorithm `Private-Centers` and the following lemma specifies its properties precisely.

**Lemma 3.1** (Algorithm `Private-Centers`)**.** *There exists an $(\varepsilon, \delta)$-differentially private algorithm such that the following holds. Assume we apply the algorithm to a database $S$ containing $n$ points in the $d$-dimensional ball $\mathcal{B}(0, \Lambda)$, with parameters $\beta, \varepsilon, \delta$. Let $P \subseteq S$ be a fixed subset (unknown to the algorithm) s.t. for a global constant $\Gamma$ we have*

$$|P| \geq \frac{\Gamma}{\varepsilon} \cdot \sqrt{d} \cdot n^{0.1} \cdot \ln\left(\frac{1}{\beta}\right) \sqrt{\ln\left(\frac{1}{\delta}\right)}.$$

*The algorithm outputs a set of at most $\varepsilon n$ centers, s.t. with probability at least $1 - \beta$ a ball of radius $O(\mathrm{diam}(P) + \frac{\Lambda}{n})$ around one of these centers contains all of $P$.*

The idea behind Algorithm `Private-Centers` is to use *locality sensitive hashing* [22] in order to isolate clustered points, and then to average clustered points with differential privacy. Algorithm `Private-Centers` captures *all* large clusters whereas the algorithm of [29] only captures *one* large cluster. We omit the proof of Lemma 3.1 due to space restrictions. In the next section we use this lemma iteratively in order to capture much smaller clusters.

### 3.1 Capturing smaller and smaller clusters

We are now ready to present the main component of our construction for the centralized model – Algorithm `Private-$k$-Means`. The algorithm privately identifies set of polynomially many candidate centers that contains a subset of $k$ candidate centers with low $k$-means cost. For readability, we have added inline comments throughout the description of `Private-$k$-Means`, which will be helpful for the analysis. These comments are not part of the algorithm. Recall that $u_1^*, \ldots, u_k^*$ denote an optimal set of centers w.r.t. the set of input points $S$, and let $S_1^*, \ldots, S_k^* \subseteq S$ denote the clusters induced by these optimal centers. (These optimal centers and clusters are *unknown* to the algorithm; they are only used in the inline comments and in the analysis.)

Throughout the execution, we use the inline comments in order to prescribe a feasible (but not necessarily optimal) assignment of the data points to (a subset of $k$ of) the current candidate centers. Specifically, we maintain an array ASSIGN, where we write $\text{ASSIGN}[j] = u$ (for some center $u$ in our current set of candidate centers) to denote that *all* of the points in the optimal cluster $S_j^*$ are assigned to the candidate center $u$. We write $\text{ASSIGN}[j] = \bot$ to denote that points in $S_j^*$ have been assigned to a center yet. For every $j$ we have that $\text{ASSIGN}[j] = \bot$ at the beginning of the execution, and that $\text{ASSIGN}[j]$ is changed exactly once during the execution, at which point the $j$th cluster is *assigned* to a center. In the analysis we argue that at the end of the execution the resulting assignment has low $k$-means cost.

**Notation.** For a point $x \in S$, we write $\text{ASSIGN}(x)$ to denote the candidate center to which $x$ is assigned at a given moment of the execution. That is, $\text{ASSIGN}(x) = \text{ASSIGN}[j]$, where $j$ is s.t. $x \in S_j^*$.

Consider the execution of the Algorithm `Private-$k$-Means`. For readability, we have summarized some of the notations that are specified in the algorithm in Table 2. We first show that the number of unassigned points reduces quickly in every iteration.

---

**Algorithm** `Private-`$k$`-Means`

---

**Input:** Database $S$ containing $n$ points in the $d$-dimensional ball $\mathcal{B}(0, \Lambda)$, failure probability $\beta$, privacy parameters $\varepsilon, \delta$.

% Let $u_1^*, \ldots, u_k^*$ denote an optimal set of centers for $S$, and let $S_j^*$ be the cluster induced by $u_j^*$, i.e., $S_j^* = \{x \in S : j = \operatorname{argmin}_\ell \|x - u_\ell^*\|\}$. For $j \in [k]$ let $r_j^* = \sqrt{\frac{2}{|S_j^*|} \sum_{x \in S_j^*} \|x - u_j^*\|^2}$, and let $P_j^* = \mathcal{B}(u_j^*, r_j^*) \cap S_j^*$.

1. Initiate $C = \emptyset$, and denote $S_1 = S$ and $n_1 = n$.

% Initiate $\texttt{ASSIGN}[j] = \bot$ for every $j \in [k]$.

2. For $i = 1$ to $\log \log n$ do

   (a) Run algorithm `Private-Centers` on the database $S_i$ with parameters $\frac{\varepsilon}{\log \log n}, \frac{\delta}{\log \log n}, \frac{\beta}{k}$, and add the returned set of centers to $C$.

   % For every $j \in [k]$: if $\texttt{ASSIGN}[j] = \bot$ and if $\exists u_j \in C$ s.t. $\|u_j - u_j^*\| \leq O(r_j^* + \frac{\Lambda}{n})$, then set $\texttt{ASSIGN}[j] = u_j$.

   (b) Let $S_{i+1} \subseteq S_i$ be a subset of $S_i$ containing $n_{i+1} = 2(T+1)wk \cdot n_i^{0.1}$ points with the largest distance to the centers in $C$, where $w = w(n, d, k, \beta, \varepsilon, \delta)$ and $T = T(n)$ will be specified in the analysis.

   % For every $j \in [k]$: if $\texttt{ASSIGN}[j] = \bot$ and if $P_j^* \not\subseteq S_{i+1}$, then let $p_j \in P_j^* \setminus S_{i+1}$, let $u_j = \operatorname{argmin}_{u \in C} \|p_j - u\|$, and set $\texttt{ASSIGN}[j] = u_j$.

3. Output $C$.

% For every $j \in [k]$: if $\texttt{ASSIGN}[j] = \bot$, then arbitrarily choose $u_j \in C$ and set $\texttt{ASSIGN}[j] = u_j$.

---

| | |
|---|---|
| $S$ | The input database. |
| $u_1^*, \ldots, u_k^* \in \mathbb{R}^d$ | An optimal set of centers for $S$. |
| $S_1^*, \ldots, S_k^* \subseteq S$ | The clusters induced by $u_1^*, \ldots, u_k^*$. |
| $r_1^*, \ldots, r_k^* \in \mathbb{R}^{\geq 0}$ | $r_j^* = \sqrt{\frac{2}{|S_j^*|} \sum_{x \in S_j^*} \|x - u_j^*\|^2}$. |
| $P_1^*, \ldots, P_k^*$ | $P_j^* = \mathcal{B}(u_j^*, r_j^*) \cap S_j^*$. |
| $S_i \subseteq S, i \in [\log \log n]$ | The set of remaining input points during the $i$th iteration. |
| $n_i = |S_i|, i \in [\log \log n]$ | The number of remaining input points during the $i$th iteration. |
| $C$ | The current set of candidate centers. |
| $\texttt{ASSIGN}[j], j \in [k]$ | The assignment constructed in the inline comments. |

**Table 2**: Notations for the analysis of algorithm `Private-`$k$`-Means`

**Claim 3.2.** *Denote $w = \frac{\Gamma \cdot \sqrt{d}}{\varepsilon} \cdot \log \log(n) \cdot \log\left(\frac{k}{\beta}\right) \sqrt{\log\left(\frac{\log \log n}{\delta}\right)}$, where $\Gamma$ is the constant from Lemma 3.1. With probability at least $1 - \beta$, for every $i \in [\log \log n]$, before Step 2b of the $i$th iteration there are at most $2kw \cdot n_i^{0.1}$ unassigned points in $S$, i.e., $|\{x \in S : \texttt{ASSIGN}(x) = \bot\}| \leq 2kw \cdot n_i^{0.1}$.*

The intuition behind Claim 3.2 is as follows. Let $S_j^* \subseteq S$ be an optimal cluster, and let $P_j^* \subseteq S_j^*$ be defined as in the first comment in the algorithm (we can think of $P_j^*$ as the subset of the $|S_j^*|/2$ points in $S_j^*$ with smallest distances to $u_j^*$). If during some iteration $i$ we have that all of $P_j^*$ is contained in our current set of input points, $S_i$, and if $|P_j^*| \geq w \cdot n_i^{0.1}$, then a center for $S_j^*$ is discovered in the $i$th iteration by the properties of `Private-Centers`. Moreover, by construction, if even a single point from $P_j^*$ is missing, then $S_j^*$ must have already been assigned to a center before the $i$th iteration. See the full version of this work for more details.

**Notation.** For $i \in [\log \log n]$ we denote by $A_i \subseteq S$ and $B_i \subseteq S$ the subset of input points whose cluster is assigned to a center during the $i$th iteration in the comments after Step 2a and after Step 2b, respectively. Observe that $A_1, B_1, \ldots, A_{\log \log n}, B_{\log \log n}$ are mutually disjoint.

Let $r_1^*, \ldots, r_k^*$ be the radii of the centers $u_1^*, \ldots, u_k^*$ as defined in the first comment in algorithm `Private-`$k$`-Means`. For a point $x \in \mathbb{R}^d$, let $u^*(x)$ denote $x$'s nearest optimal center, and $r^*(x)$ its corresponding radius. The next observation is immediate from the construction.

**Observation 3.3.** *For every $i \in [\log \log n]$ and for every $x \in A_i$, at the end of the execution we have*

$$\|x - \text{ASSIGN}(x)\|^2 \leq O\left(\|x - u^*(x)\|^2 + (r^*(x))^2 + \frac{\Lambda^2}{n^2}\right).$$

We charge the cost of points $x \in B_i$ to points that were already assigned to centers in some iteration $j \leq i$ in the sense specified by the following lemma.

**Lemma 3.4.** *With probability at least $1 - \beta$, for every iteration $i \in [\log \log n]$ and for every $x \in B_i$ there exists a set of input points $Q(x) \subseteq S$ such that*

1. *For every $i \in [\log \log n]$ and for every $x \in B_i$ it holds that $|Q(x)| = T$, where $T = O(\log \log n)$.*

2. *For every $i \in [\log \log n]$ and for every $x, y \in B_i$, if $x \neq y$ then $Q(x) \cap Q(y) = \emptyset$.*

3. *For every $i \in [\log \log n]$ and for every $x \in B_i$, at the end of the execution it holds that*

$$\|x - \text{ASSIGN}(x)\|^2 \leq O\left(\|x - u^*(x)\|^2 + (r^*(x))^2 + \frac{1}{T}\sum_{q \in Q(x)} \|q - \text{ASSIGN}(q)\|^2\right).$$

Intuitively, Lemma 3.4 follows from the fact in every iteration $i$, for every *unassigned* point in $S$ there are at least $T$ *assigned* points in $S_i$. We omit the proof due to space restrictions.

**Lemma 3.5.** *If Algorithm `Private-k-Means` is applied to a database $S$ containing $n$ points in the $d$-dimensional ball $\mathcal{B}(0, \Lambda)$, then it outputs a set $C$ of at most $\varepsilon n \log(\frac{k}{\beta})$ centers, s.t. with probability at least $1 - \beta$*

$$\text{OPT}_S(C) = \min_{\substack{D \subseteq C \\ |D| = k}} \{\text{cost}_S(D)\} \leq O(1) \cdot \text{OPT}_S + O\left((Twk)^{1.12}\right) \cdot \Lambda^2,$$

*where $w$ is defined in Claim 3.2, and $T = \Theta(\log \log n)$. The exponent $1.12$ is arbitrary and can be reduced to any constant $a > 1$.*

*Proof.* We show that the stated bound holds for the assignment described in the inline comments throughout the algorithm (the array `ASSIGN`) at the end of the execution. First observe that by Claim 3.2 and by the fact that there are $\log \log n$ iterations, at the end of the execution there could be at most $O\left((2(T+1)wk)^{1.12}\right)$ unassigned input points. Let us denote the set of unassigned points as $H$. The distance from each unassigned point to an arbitrary center is trivially at most $\Lambda$. For every assigned point $x$, by Observation 3.3 and by Lemma 3.4, either $\|x - \text{ASSIGN}(x)\|^2 = O(\|x - u^*(x)\|^2 + (r^*(x))^2 + \frac{\Lambda^2}{n^2})$, or

$$\|x - \text{ASSIGN}(x)\|^2 \leq O\left(\|x - u^*(x)\|^2 + (r^*(x))^2 + \frac{1}{T}\sum_{q \in Q(x)} \|q - \text{ASSIGN}(q)\|^2\right).$$

Hence,

$$\text{cost}_S\left(\{\text{ASSIGN[j]} : j \in [k]\}\right) = \sum_{x \in S} \|x - \text{ASSIGN}(x)\|^2$$

$$= \sum_{x \in H} \|x - \text{ASSIGN}(x)\|^2 + \sum_{\substack{i \in [\log \log n] \\ x \in A_i}} \|x - \text{ASSIGN}(x)\|^2 + \sum_{\substack{i \in [\log \log n] \\ x \in B_i}} \|x - \text{ASSIGN}(x)\|^2$$

$$\leq O\left((2(T+1)wk)^{1.12}\right) \cdot \Lambda^2 + \sum_{\substack{i \in [\log\log n] \\ x \in A_i}} O\left(\|x - u^*(x)\|^2 + (r^*(x))^2 + \frac{\Lambda^2}{n^2}\right)$$

$$+ \sum_{\substack{i \in [\log\log n] \\ x \in B_i}} O\left(\|x - u^*(x)\|^2 + (r^*(x))^2 + \frac{1}{T}\sum_{q \in Q(x)} \|q - \mathtt{ASSIGN}(q)\|^2\right)$$

$$\leq O\left((2(T+1)wk)^{1.12}\right) \cdot \Lambda^2 + \sum_{x \in S} O\left(\|x - u^*(x)\|^2 + (r^*(x))^2\right)$$

$$+ \frac{1}{T}\sum_{\substack{i \in [\log\log n] \\ x \in B_i \\ q \in Q(x)}} O\left(\|q - \mathtt{ASSIGN}(q)\|^2\right)$$

$$\leq O\left((2(T+1)wk)^{1.12}\right) \cdot \Lambda^2 + O(1) \cdot \mathrm{OPT}_S + \frac{1}{T}\sum_{\substack{i \in [\log\log n] \\ x \in B_i \\ q \in Q(x)}} O\left(\|q - \mathtt{ASSIGN}(q)\|^2\right) \quad (1)$$

Now recall that for every $i \in [\log\log n]$ and for every $x \neq y \in B_i$ it holds that $Q(x) \cap Q(y) = \emptyset$. Hence, every point $q \in S$ contributes at most $\log\log n$ times to the last summation above. So,

$$(1) \leq O\left((2(T+1)wk)^{1.12}\right) \cdot \Lambda^2 + O(1) \cdot \mathrm{OPT}_S + \frac{\log\log n}{T}\sum_{q \in S} O\left(\|q - \mathtt{ASSIGN}(q)\|^2\right)$$

For $T = \Theta(\log\log n)$ (large enough) we get that the last term above is at most half of the left hand side of the inequality, and hence,

$$\mathrm{cost}_S\left(\{\mathtt{ASSIGN[j]} : j \in [k]\}\right) \leq O\left((2(T+1)wk)^{1.12}\right) \cdot \Lambda^2 + O(1) \cdot \mathrm{OPT}_S$$

$\square$

**Lemma 3.6.** *Algorithm `Private-k-Means` is $(\varepsilon, \delta)$-differentially private.*

The privacy analysis of Algorithm `Private-k-Means` is standard, and is omitted due to space restrictions. Intuitively, in every iteration, Step 2a satisfies differential privacy by the properties of Algorithm `Private-Centers`, and we use the following technique for arguing about Step 2b: Let $X$ be an ordered data domain and let $\mathcal{A}$ be a differentially private algorithm that operates on a multiset of $m$ elements from $X$. Then for any $n \geq m$, the algorithm that takes a multiset $S$ of $n$ elements from $X$ and runs $\mathcal{A}$ on the smallest (or largest) $m$ elements in $S$ is differentially private. The intuition is that changing at most one element in $S$ can change at most one element of the multiset that we give to $\mathcal{A}$, and this change is "hidden" by the privacy properties of $\mathcal{A}$. See [9] for more details and applications of this technique.

Combining Lemmas 3.5 and 3.6 with Theorem 2.1 yields the following theorem.

**Theorem 3.7.** *There is an $(\varepsilon, \delta)$-differentially private algorithm that, given a database $S$ containing $n$ points in the $d$-dimensional ball $\mathcal{B}(0, \Lambda)$, identifies with probability $1 - \beta$ a $(\gamma, \eta)$-approximation for the $k$-means of $S$, where $\gamma = O(1)$ and $\eta = \mathrm{poly}\left(\log(n), \log(\frac{1}{\beta}), \log(\frac{1}{\delta}), d, \frac{1}{\varepsilon}, k\right) \cdot \Lambda^2$.*

**Acknowledgments.** We would like to thank Moni Naor for helpful discussions, and the anonymous reviewers for useful suggestions and corrections.

## Footnotes

[3]The algorithm of [18] obtains $O(1)$ multiplicative error and $\tilde{O}(k^2 d)$ additive error.

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
