[Reviews · NeurIPS 2018]

Reviewer 1



The paper considers k-means under differential privacy. The main idea of differential privacy (applied to k-means) is that changing one point in the input should not significantly change the centers that are computed. This immediately means that any approximation algorithm for the differentially private k-means problem needs to have additive error: The introduction includes a simple example of points in k locations (=> optimum has zero cost). Now when moving one of these points to a different location, the centers may not change significantly; ergo one of the solutions has to have a large additive error; basically an error of D^2 where D is the largest pairwise distance. Thus, a common approach for this problem is to compute an approximation that is both multiplicative and additive. The best known result prior to this paper achieves a quality of O(k)*OPT + O~(d^0.51+k^1.51). The paper dramatically reduces this guarantee to O(1)*OPT + O~(k^1.01 d^0.51 + k^1.5). The new algorithm works both in the centralized variant where a trusted server provides anonymization, and in the local variant, where the users themselves randomly change their point before sending it to an untrusted server. The results look very good to me; better multiplicative approximation, no increase in the additive term, and in the local model, the number of rounds is also decreased. My only concern is whether the paper really fits NIPS, because a) not very much of the proof makes it into the actual paper, and b) there are no experiments. It's not clear to me how practical the algorithm is; the main routine does not look to complicated, but the subroutine might be. Aside from the missing experiments, this is clearly the best paper in my (not necessarily representative) batch. I do not know whether a paper without experiments can appear at NIPS; I'll vote for acceptance and wait for the other reviewer's opinion on that. The introduction is very well written. The problem definition becomes clear immediately, and the need for additive approximation is explained nicely. The best non-private approximation for k-means is cited. I'm no expert on differentially private k-means algorithms, but the related work section wrt that topic looks solid to me as well. I appreciated the paragraph about classical algorithm not being private, and the pointer to the k-means++ adaptation. The technical contribution is also made very clear; previous work is summarized clearly. The paper builds upon a previously obtained result which essentially gives a differentially private algorithm for k-means if and only if one can supplement it with a set of good center candidates of poly(n) size. At first glance, this seems easy, since for non-private k-means, one can simply take the input points as the candidate set, this is of size n and has an error of 2 compared to centers from R^d. However, for private k-means clustering, this is not possible -- the centers can not be a subset of the input points, because otherwise, removing a point may significantly change the computed centers. The paper has two main contributions. First, it shows how to compute a good center set; this part is based on a different related paper, but seems refined. Second, it embeds both the candidate set computation and the subroutine that computes good centers given such a candidate set into a larger main algorithm. Roughly, this algorithm computes a candidate set, uses it to compute centers, and then eliminates points which it now deems "covered" by the new centers. This process is then repeated log log n times, and then the probability that all points are covered is sufficiently high. A question for the author feedback: Just for my clarification: How does the fact that we can not pick S as the candidate center set show in Theorem 2.1? What if I just hand the algorithm in [16,6] Y=S, how does reading the theorem tell me that that does not lead to a differentially private algorithm? Smaller remarks: - In Table 1, the ~ notation hides log n - factors. As far as I thought, the O~(X) is usually used to hide factors that are polylogarithmic in X -- but n is not a part of the expression here. It is stated in the caption of Table 1 that logarithmic factors in n are omitted; yet I think the dependence on log n should be stated somewhere in the introduction in the text (if I missed it, sorry). - The last sentence on page 3 (line 112+113) confused me quite a bit. It might be a good point in the paper to describe that one can not simply take the input points as a candidate set. This is obvious once observed, but when usually working on non-private k-means, the statement that an O(1)-approximation would work for a discrete set and is simply not applicable in R^d just sounds strange. - The citations for non-private coresets in line 178 seem a bit arbitrary; they should probably contain at least Dan Feldman, Michael Langberg: A unified framework for approximating and clustering data. STOC 2011: 569-578 which is to my knowledge the currently best for k-means. - Theorem 2.1: "a finite set centers" -> "a finite set of centers" - The sentence above Lemma 3.5 that the proof is omitted seems to be not true, since the proof is there. Edit after author feedback: I like this submission, and this has not changed.

Reviewer 2



This paper considers the problem of differentially private k-means clustering, where the input to the algorithm is a collection of n points belonging to the ball B(0,Lambda) in d dimensions, and the goal is to output a set of k centers minimizing the k-means cost, while at the same time preserving (epsilon, delta)-differential privacy. The goal is to achieve utility guarantees ensuring that, with high probability, the cost of the output clustering is at most gamma * OPT + eta, where OPT is the minimum possible k-means cost. The main result in this paper is an algorithm for which gamma is constant, while in all prior work gamma scales with the problem parameters such as the dimension, target number of clusters, or dataset size. The results from this paper also extend to local differential privacy, and enable the construction of coresets. The paper builds on techniques from prior work. In particular, the algorithm follows a two step procedure where first a set of candidate centers is obtained (which is a relatively small set of points that should contain a set of k good centers). Second, a greedy swapping algorithm is used to actually find a good set of centers from the candidate set. This basic scheme was used previously by Balcan et al. (2017), and the main technical contribution is a new candidate center construction algorithm. The proposed construction builds on the work of Nissim and Stemmer. The construction has two components: first, a routine called Private-Centers that uses locality sensitive hashing that is able to identify good cluster centers for all sufficiently large clusters (relative to the size of the dataset). To obtain good candidate centers for all clusters (even small ones), the Private-Centers procedure is repeatedly applied to the set of points that are furthest from the so-far chosen candidate set. Overall the paper is clearly written, intuition is provided for the main theorems, and the results are interesting. It would be nice to include a few more details about the private coreset constructions and the utility guarantees in the local model. One thing that may be worth clarifying is why step (2b) of Private-k-means preserves privacy, which (as I understand it) depends on the fact that the set of candidate centers C chosen so far was picked while preserving privacy.

Reviewer 3



This paper studies k-means clustering in Euclidean spaces under differential privacy constraints. In the classical DP setup, the result improves all existing works in this field, obtaining constant multiplicative factor and $poly(k,d,\log n,1/eps)$ additive error. This resolves an important open question in this field and closes a long line of research in pursuit of private clustering algorithm. The paper also establishes results for the local DP setup, and gets $O(n^{2/3})$ additive error with $O(1)$ rounds of communication. I think this is a very interesting paper with solid contribution in this field. The algorithm follows the strategy by Balcan et al.,(2017): a candidate set of centers is first constructed in a private way, on which they can run private clustering algorithm in discrete space by Gupta et al. The previous approach gives $O(\log^3 n)$ approximation ratio due to the mismatch between hierarchical partition and Euclidean norm. The approach proposed in this paper overcomes this difficulty by exploring the data points at different scales using non-hierarchical partitions by locality sensitive hashing. I would like to see this paper published on NIPS advancing the frontiers in this area. The claimed contribution for the core-set part is not well-justified, however. It is known that cluster centers with appropriate weights themselves serve as constant-factor core-set, so it's not some new contribution in addition to the clustering part. Actually, the open problem proposed by Feldman et al., (2009) is about (1+a) approximation for arbitrarily small a>0.